# A New Lineage of Perch Rhabdovirus Associated with Mortalities of Farmed Perch

**DOI:** 10.3390/pathogens10101256

**Published:** 2021-09-28

**Authors:** Laurane Pallandre, Armand Lautraite, Claudette Feuvrier, Françoise Pozet, Laurent Dacheux, Laurent Bigarré

**Affiliations:** 1Laboratory of Ploufragan-Plouzané-Niort, ANSES, Technopole Brest Iroise, 29280 Plouzané, France; laurane.pallandre@anses.fr; 2Private Veterinarian, 82170 Grisolles, France; armand.lautraite@gmail.com; 3Jura Departemental Analysis Laboratory, 39802 Poligny, France; cfeuvrier@jura.fr (C.F.); fpozet@jura.fr (F.P.); 4Institut Pasteur, Université de Paris, Lyssavirus, Epidemiology and Neuropathology Unit, 28 rue du Docteur Roux, CEDEX 15, 75724 Paris, France; laurent.dacheux@pasteur.fr

**Keywords:** *Rhabdoviridae*, *Perhabdovirus*, fish, percid, outbreak

## Abstract

A perhabdovirus was isolated from a mortality episode affecting a fish farm in 2019 in Western Europe. This virus was produced in cell culture and was readily detected by a species-specific real-time PCR assay. The near-complete sequence of the virus obtained showed some relatedness with viruses of the species *Perhabdovirus perca*. However, it was distinct enough from these viruses to form a separate genetic lineage. Multiple substitutions along the genome caused non-detection using a range of conventional PCRs previously shown to target four known genogroups of perhabdoviruses. However, various generic PCRs efficiently detected the isolated virus. The origin of this virus remains to be elucidated. It may have been introduced into the farm via wild genitors. This finding provides new evidence of the high genetic diversity of percid perhabdoviruses and the potential of new genotypes to emerge as threats for fish farming. Efforts to improve the existing diagnostic methods and control this large group of viruses are still needed.

## 1. Introduction

Over the past four decades, a group of perhabdoviruses (family *Rhabdoviridae*) have caused losses of percid (perch and pike-perch) fry, juveniles and adults in European countries, mostly on farms and in experimental facilities [1,2,3,4,5,6,7,8,9,10,11]. Sick fish usually display abnormal swimming behavior, such as swirling, alternating with lethargy. In some cases, hemorrhaging at the base of the fins can occur. 

The *Perhabdovirus* genome is made up of a linear single-stranded negative RNA of about 11.5 kb encoding the five canonical rhabdovirus genes in the following order: nucleoprotein (N), polymerase-associated phosphoprotein (P), matrix protein (M), glycoprotein (G) and RNA-dependent RNA polymerase (L). For some viruses, additional small open reading frames (ORFs) are encoded, but there are no experimental data illustrating their expression or role in the molecular biology of the virus. The genus *Perhabdovirus* includes four viral species recognized by the International Committee on Taxonomy of Viruses (ICTV): *Perhabdovirus anguilla* represented by viruses infecting eels only (i.e., EVEX), *Perhabdovirus perca* (formerly *Perch perhabdovirus*) and *Perhabdovirus trutta* (formerly *Sea trout perhabdovirus*), both represented by diverse viruses infecting percids as well as a few other fish species *(*i.e., Sea trout rhabdovirus and virus R6146), and *Perhabdovirus leman* (including Leman rhabdovirus (LeRV)), described as virus 18-193 [12]. Genomic data have demonstrated the existence of four main genogroups infecting percids. One genogroup is composed of all the viruses belonging to *Perhabdovirus perca*. A second cluster includes members of *Perhabdovirus trutta*. A third cluster is composed of a single virus (LeRV) representing *Perhabdovirus leman*. A fourth cluster includes a pair of viruses from France (18-203 and 18-206) that were initially thought to represent a new species considering the limited levels of nucleic acid identities of the concatenated genes with the closest virus R6146 (80%). However, considering the new rules of demarcation of ICTV for perhabdoviruses, viruses 18-203 and 18-206 should be classified as members of the *Perhabdovirus trutta* species since the levels of identity of their amino acid sequences with some members of this taxon are superior to 90%.

Considering this genetic diversity, four sets of primers have previously been designed, each one able to amplify a specific portion of the N gene of all the members of a given genogroup [12]. Although the specificity of these assays is not high enough for diagnostic tests, because they occasionally produce non-specific products, they are useful as an N-based identification tool for any new isolate produced in cell culture or directly from infected fish.

Recently, accurate genotyping methods have proved powerful for tracing some of the outbreaks of percid perhabdoviruses in Europe [7]. Nevertheless, molecular data are still scarce across the continent. Wild fishes, not only percids, represent a reservoir for a number of perhabdoviruses, known and unknown, which are regularly introduced into farms or experimental facilities and are then disseminated between farms over Europe via international trade (eggs, fry, genitors). Appropriate prophylactic measures, such as egg disinfection, can help prevent the vertical transmission of these viruses, but they are not consistently employed by farmers. More generally, efficient diagnostic methods also lack for some of these viruses, and the few existing methods are insufficiently applied in European laboratories. 

In 2019, an outbreak occurred in perch larvae on a farm in Western Europe, reaching a mortality rate of nearly 30%. From sick fish, we isolated a perhabdovirus and obtained a near-complete sequence, which demonstrated the presence of a new genotype hitherto undescribed belonging to the *Perhabdovirus perca* species.

## 2. Results

### 2.1. Pathology

In December 2019, perch juveniles maintained in tanks in freshwater were affected by morbidities and mortalities. The mortality rate was 0.3 to 1.5% daily, reaching about 30% at the end of the episode. A batch of 50 individuals was examined. All fishes had open gills and about 50% exhibited spinal distortion (Figure 1). Abnormal swimming (swirling) was common. Several bacterial species were isolated, namely *Aeromonas* sp., *Pseudomonas fluorescens* and *P. alcaligenes* in the kidney, as well as *Aeromonas* sp., *P. fluorescens* and *Providencia stuartii* in the brain. No flavobacteriia were detected and no parasites were observed either in the gills, the skin, the fins or in the intestine. 

### 2.2. Virus Detection

Seven days after inoculation of two cell lines with the supernatant of 10 fish ground together, a cytopathic effect typical of that induced by perhabdoviruses was observed. The cell layer was progressively destructed. After a second passage of the supernatant of these, a strong ECP appeared only 24 h after inoculation. In another assay, after freezing at −80 °C for long-term storage, a second passage led to an ECP as soon as 4.5 dpi with a complete destruction of the cells at 5.5 dpi (Appendix A). Considering the host species affected by the disease and the symptoms (swirling), the presence of a perhabdovirus was suspected. 

This viral isolate was named 20-43. Therefore, a series of specific RT-PCR assays was performed starting from nucleic acids extracted from the supernatants of cell cultures or directly from the ground organs. A real-time RT-PCR assay targeting viruses of the *Perhabdovirus perca* species produced signals on both nucleic acid extracts, with Ct values between 20.7 and 27.3 from cell culture supernatants and organ extracts, respectively (Figure 2). However, the maximum levels of fluorescence of the amplification curves were lower than those obtained with a positive control, which exhibited a higher Ct value (31.9) and about two-fold the final fluorescence level compared with the isolate 20-43. This result suggested the presence of a perhabdovirus related to the tested genogroup, but its sequence likely had some mismatches that affected the signal obtained with the real-time RT-PCR primers and probe. 

It must be mentioned that, a few months after the mortality episode affecting young juveniles, three individual healthy fish, older than the ones tested above and maintained in a different site, were tested negative both by cell culture and real-time RT-PCR assays (not shown). Furthermore, in 2021, five healthy individual juveniles, from the progeny of genitors originating from the farm, were also tested negative by real-time RT-PCR. Therefore, healthy fish from two different generations were negative for this virus.

To genotype the virus, a range of conventional RT-PCR assays targeting the nucleoprotein genes of viruses from the four known genogroups was performed as previously described. Surprisingly, no signal was produced with any of the four RT-PCR assays, suggesting the virus 20-43 was relatively different from the known perhabdoviruses (not shown). However, an experimental generic cPCR targeting conserved portions of the 3′ end of the percid perhabdoviruses and a fragment of the N ORF produced a clear signal of the expected size (Figure 2). A second generic assay (oPVP141/143) targeting the 3′ end of the N gene was also positive (Figure 2). Similarly, two cPCR assays targeting a variable fragment of the 5′ end of the genome of percid perhabdoviruses each produced a signal of size identical to the one expected for perch rhabdoviruses (Figure 2). However, for one particular assay (primer pair oPVP705/706), the size of the product was different from the size of the amplicons obtained with very distinct viral genotypes (Figure 2). Finally, two generic RT-PCR assays targeting either a large portion of the G gene or the 3′ half of the N gene, also produced bands at the expected sizes. All the sequences of these diverse genetic fragments confirmed the presence of a virus related to, though distinct from, perch rhabdovirus. These RT-PCR assays completed the missing portions of both N and G by combining strain-specific and generic primers (Figure 2). In summary, using primers amplifying different regions of perhabdoviruses, a total of seven amplicons were obtained whose sequences led to the complete N and G ORFs. 

To understand why the genome of virus 20-43 was not detected using cRT-PCR with any of the four sets of genogroup-specific primers, the primer sequences were aligned with the sequence of the N ORF newly obtained from virus 20-43. All primers exhibited several mismatches with their homologous sequence in virus 20-43. For instance, there were a total of six mismatches between the N ORF of virus 20-43 and primers oPVP546 and oPVP547 targeting *Perhabdovirus* members. These numerous mismatches explained the absence of amplification signals for all the cRT-PCR primers. However, there were also some mismatches between the N ORF and the oligonucleotides used for real-time RT-PCR: two to three for the primers, and two for the TaqMan probe. It was, therefore, surprising that a signal was produced using real-time RT-PCR, although the maximum fluorescence level of the amplification curve was lower than the one of a positive control perfectly adapted to the probe and primers.

### 2.3. Genome Sequence

By NGS sequencing the present viral isolate 20-43, a sequence of 11,419 nt was obtained. This sequence did not contain the extremities of the genome but encompassed all the complete virus genes. As previously mentioned, a RT-cPCR assay targeting the 3′ end of the genome led to the production of an amplicon, which once sequenced, added 15 nt to the genomic sequence extending it to 11,434 nt. In comparison with the ends of the genomes of the most related viruses, it was speculated that 3 nt (possibly CGT) were missing at the 5′ end of perhabdovirus 20-43, as well as 20 nt at the 3′ end, leading to an estimated size of 11,457 nt for virus 20-43. If correct, this size is similar to those of the three sequenced members of the *Perhabdovirus perca* species and about 150 nt shorter than those of viruses R6146, 18-193 and 18-203.

The genetic organization of virus 20-43 was typical of perhabdoviruses, with the five canonical genes N-M-P-G-L (Figure 2). No additional minor ORFs (threshold 180 nt) were detected. The typical translation signal AACAG was present upstream of each ORF, and the termination transcription/polyadenylation (TTP) signal TATGA (7) was perfectly conserved downstream each ORF. Compared with related viruses from percids, the N ORF of virus 20-43 was slightly longer with a trinucleotide (CTG, Leucine) inserted at the end, just before the stop codon. A trinucleotide insertion was also observed in the P ORF (963 nt) compared with the closest viruses members of *Perhabdovirus perca* 16-65, 4890 and Perch rhabdovirus (960 nt). However, the P ORF was much shorter than its homologues from more distant viruses of the genus, for instance R6146 (993 nt). The M ORF had exactly the same size (657 nt) as viruses P8350 and 16-65, but it was slightly different from the corresponding ORF of other viruses such as LeRV (660 nt) and R6146 (642 nt). The length of the L ORF was strictly similar to that of the most related viruses from percids.

### 2.4. Phylogenetic Studies

The complete nucleotide sequence of the N ORF of virus 20-43 was aligned with a set of homologues belonging to the genus *Perhabdovirus*. It shared 69 to 81% of its nucleic acid identity with viruses from the genus *Perhabdovirus*, excluding EVEX which was more distant (Figure 3A). The most related viruses were those belonging to *Perhabdovirus perca*, for instance French viruses M7173, 18-200 and 18-195 as observed in the phylogenetic tree (Figure 3A). The N protein had a level of amino acid sequence identity of 89–92% with the N of viruses belonging to *Perhabdovirus perca* and less than 90% with the N of other perhabdoviruses (Figure 3B). Considering the L ORF, the amino acid sequence identity of virus 20-43 varied between 80 and 91% identity with its homologues from other percid perhabdoviruses and identity only reached 72% with the L ORF of EVEX (Figure 3B). 

The concatenated nucleotide ORFs sequences were also aligned with the only six concatenated nucleotide ORFs sequences available from other related percid perhabdoviruses (Figure 4A). The identity levels of nucleic acids were 72–78% and 71–78%, respectively, with these six most related viruses. In parallel, we also aligned the complete nucleotide L ORF sequences alone and demonstrated the strong contribution of the L gene in the variability of the coding genome since the levels of identities of the L and the concatenated ORFs were similar (Figure 4A). The phylogenetic tree, based on the concatenated ORF sequences of these different viruses (N, P, M, G and L; 10107 positions compared) confirmed that virus 20-43 was included the genus *Perhabdovirus* together with other rhabdoviruses from percids (Figure 4B). More precisely, all these data indicate that the virus 20-43 belongs to the species *Perhabdovirus perca* (formerly *Perch perhabdovirus*). The four recognized species of the *Perhabdovirus* genus were within a clade distinct to another clade grouping rhabdoviruses from dolphin, tortoise or various fish species (Figure 4B). 

## 3. Discussion

A virus related to perhabdoviruses was isolated from perch juveniles affected by a mortality event on a European fish farm at the end of year 2019. This virus was replicated in cell culture and was readily detected using a set of generic cRT-PCR assays and a species-specific real-time RT-PCR assay. However, the virus was not detected using more specific assays targeting the four known genogroups of percid perhabdoviruses. This absence of detection was *a posteriori* explained by multiple mismatches between the viral sequence and the oligonucleotides. Its complete sequence confirmed its relationship to the other percid perhabdoviruses and its closest identities with viruses of the species *Perhabdovirus perca*, though there were numerous substitutions compared with the members of this species and with viruses from other species. Despite these nucleotide changes and regarding the threshold of protein identity necessary to delineate two species (90%), we consider virus 20-43 as a member of *Perhabdovirus perca*, which is the most represented species in the outbreaks over the past decades. However, perhabdovirus 20-43 seems distinct enough to represent a new viral lineage within the species. 

The origin of perhabdovirus 20-43 is unknown, as is its mode of introduction into the farm. The farm is supplied with water from a spring, which cannot have been the source of the virus. Interestingly, several months before this episode, another mortality event affecting perch occurred. It happened just after connecting the water circulation system of tanks containing a few genitors captured in the wild to tanks containing fish from the farm. These latter fish were soon affected by morbidity and mortality. Unfortunately, no analysis was performed on the dead fish from this first episode. Hence, there is a possibility that the wild fish were carriers of the virus and transmitted it horizontally to the perch produced in the farm via the water or fomites.

Regarding the mortality observed in December 2019 in this Western European fish farm, it affected fish just after their transfer to new tanks to start a new growing phase. Diverse bacteria were found in the dead fish. However, they are generally considered as opportunistic and, therefore, likely not responsible of the mortality. A hypothesis is that virus 20-43 was at least partly responsible for the disease. The larvae were perhaps already infected with the virus, albeit without any associated pathology. The manifestation of the infection may have been triggered by either a change in rearing conditions that subjected the fish to stress, therefore, making them more vulnerable, or, less likely, a sudden evolution of the virulence of this virus. From other episodes on other farms in Europe, it is known that the weaning of larvae and a change in the feeding regime induces stress that can trigger viral replication. Generally, the pathology of percid perhabdoviruses has been poorly studied and it is not clear why some members cause massive mortalities to larvae and others cause only morbidity to juveniles and adults. More research is needed to understand the virulence mechanisms of this viral genus. 

A comprehensive overview of the diversity of the perhabdoviruses is needed to prevent their spread across Europe. The present genetic data as well as amplification methods can contribute to the rapid and accurate identification of any new isolate in order to compare it with known isolates, which is crucial for tracing outbreaks. Due to the general lack of active surveillance in Europe, it is likely that known, and yet unknown, genotypes and species will emerge in the next decade [12]. Furthermore, given the host-switching capacity of these viruses and their frequent geographic translocations, the emergence of any of these viruses will likely affect various hosts—not only those already recognized (percids and non-percids), but also possibly new freshwater hosts that are increasingly farmed worldwide, such as tilapias.

## 4. Materials and Methods

### 4.1. Virus Isolation

Samples were isolated in December 2019 from symptomatic *Perca fluviatilis* juveniles (size 3.3 cm; weight 0.4 g) maintained in freshwater at 16–18 °C. The supernatant of 10 fish ground together was used to inoculate two cultured cell lines, BF2 and RTG2 (ECACC, MERCK-SIGMA), as already described [12]. Several months after this mortality episode, in June 2020, other fish from an older generation, healthy and originating from the same farm, were tested for the presence of perhabdoviruses (Appendix A). This batch was used as genitors to provide a new generation of juveniles. Five of these juveniles, all healthy, were also submitted to a specific diagnostic test in September 2021.

### 4.2. Nucleic Acid Extraction, RT-PCR Detection and Sanger Sequencing

Total nucleic acids were extracted using a NucleoSpin Virus kit (Macherey-Nagel, Düren, Germany) with minor modifications as previously described [12]. The presence of a perhabdovirus was initially tested using a newly designed generic real-time RT-PCR assay detecting various viruses of the *Perhabdovirus perca* species (Figure 1). The pair of primers oPVP529 (GTGCAGGAARTCACCATACTCATC), oPVP530 (CAGCAGAGCACAGGTCATTTG) and a TaqMan probe, TqPVP30 (FAM-TCTGCCAATCCTGGAGTTCACTGCTG-BHQ1), were used at 0.4 µM in a final reaction volume of 25 µL containing 5 µL of total nucleic acids. A SuperScript III One-Step qRT-PCR kit (Invitrogen) was used with the following thermo-cycling: a reverse-transcription step of 30 min at 50 °C, followed by 15 min at 95 °C and 40 cycles of 15 sec at 94 °C and 60 sec at 60 °C. A viral isolate (16-121) highly related to Perch rhabdovirus was used as a positive control [7]. 

Further molecular tests were performed using a range of RT-conventional PCRs (RT-cPCRs) combining generic and strain-specific primers (Figure 1 and Appendix A). For all the amplification reactions, a SuperScript III One-Step with Platinum Taq High-Fidelity kit (Invitrogen) was used according to the manufacturer’s protocol by adapting the polymerization time to the length of the expected product and the melting temperature of the primers. Two overlapping fragments of the G gene were amplified as previously described [2]. Similarly, the complete N ORF was obtained by producing different amplicons, generic or strain-specific [2]. The 3′ end of the genome was amplified using one primer (oPVP714) targeting a region that is highly conserved among all the percid perhabdoviruses and another primer (oPVP717) targeting a moderately conserved region of N. The produced amplicon of 521 bp contained a part of the N ORF and a major part of the non-translated sequence upstream. The 5′ end of the viral genome was amplified using the two pairs of primers already published: oPVP705/oPVP706 and oPVP706/oPVP707 [12]. 

After RT-PCR, 10 μl of these reactions was loaded on a 2% agarose E-Gel (Invitrogen) and migrated 15 min before observation under UV light. Before sequencing, PCR products were purified using a NucleoSpin PCR clean-up kit (Macherey-Nagel) and subsequently, TA-cloned in a PCR4-TOPO vector (Invitrogen). For each amplicon, three clones were sequenced in both orientations with universal primers using the Sanger method and a 3130 Genetic Analyzer (Applied Biosystems).

### 4.3. Next-Generation Sequencing

To obtain the near-complete genome of the virus, next-generation sequencing (NGS) was performed using Illumina sequencing technology as previously described [12,13]. For the very few regions with low covering or uncertainties, Sanger sequencing was performed after specific amplifications by RT-PCR. The complete viral genome sequence is available in GenBank (MW685822).

### 4.4. Phylogenetic Analysis

DNA sequences obtained from Sanger sequencing were assembled and the consensus sequence edited using VectorNTI11.5 (Invitrogen). The identities levels were obtained by aligning the sequences using the ‘align’ function of VectorNTI and the identity matrix was constructed using the RStudio software. For the phylogenetic analyses, the sequences were aligned with the ClustalW function of MEGA7 or MEGA10 [14,15]. Maximum-likelihood (ML) analyses were performed using a transition-to-transversion ratio of 2.0 with empirical base frequencies and the HKY85 substitution model implemented in the MEGA program [14]. Bootstrap analyses were performed using 1000 replications. Values greater than 90% were considered strong evidence for robust phylogenetic groupings.

## Figures and Tables

**Figure 1 pathogens-10-01256-f001:**
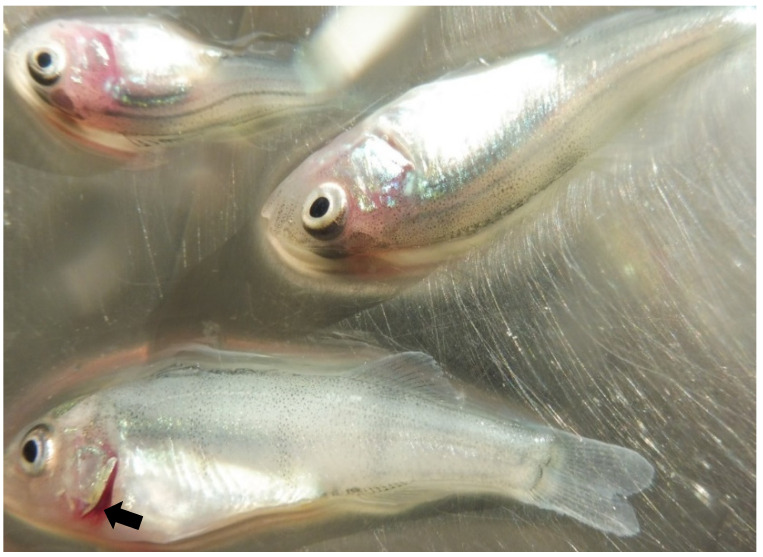
*Perca fluviatilis* juveniles affected by a mortality episode in a fish farm from Western Europe in 2019. The arrow shows the open gills observed in affected animals.

**Figure 2 pathogens-10-01256-f002:**
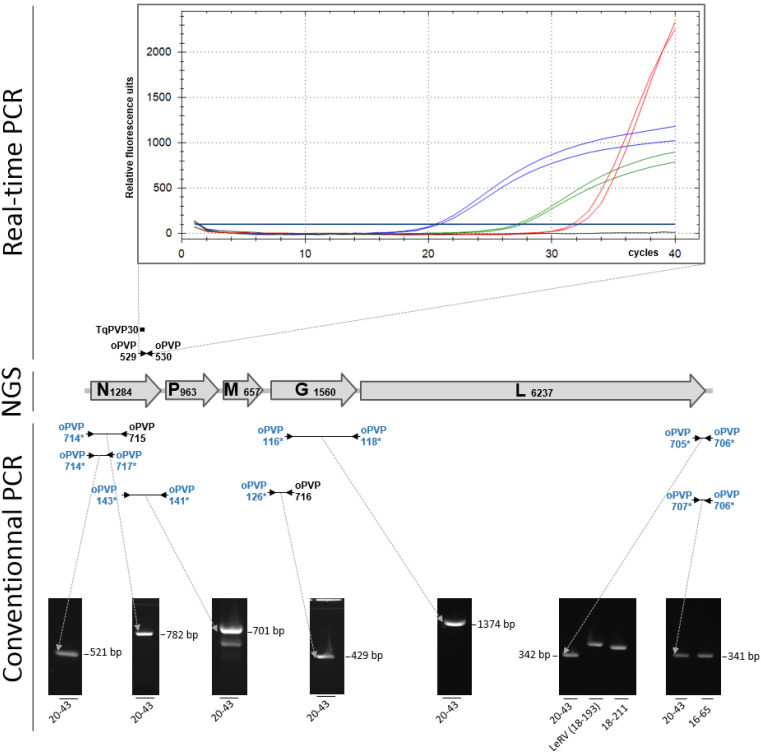
Molecular detection and genetic characterization of virus 20-43. The near-complete sequence of the viral genome was obtained by combining next-generation sequencing (NGS) and conventional PCR (cPCR) (see text). ORFs and their size (in nt) are indicated on the map. A real-time RT-PCR targeting viruses belonging to the *Perhabdovirus perca* species gave curves showing an unusual shape compared with controls (blue lines, curves obtained from cell cultures infected with 20-43; green lines, curves obtained from infected fish organs; red lines, curves of a positive control represented by a strain of perch perhabdovirus); the three samples are in duplicate. Various cPCRs (dark lines) were used to confirm the presence of a perhabdovirus in the inoculated cell culture. Primers are symbolized by arrowheads. Generic primers (in blue) targeting diverse percid perhabdoviruses are labelled with asterisks; specific primers designed from the sequence of virus 20-43 are in black. Three related perhabdoviruses were used as controls for the amplifications of a region in the 5′ end of the genome [12]. Note the different sizes of the amplicons for viruses 18-193 and 18-211 compared to virus 20-43.

**Figure 3 pathogens-10-01256-f003:**
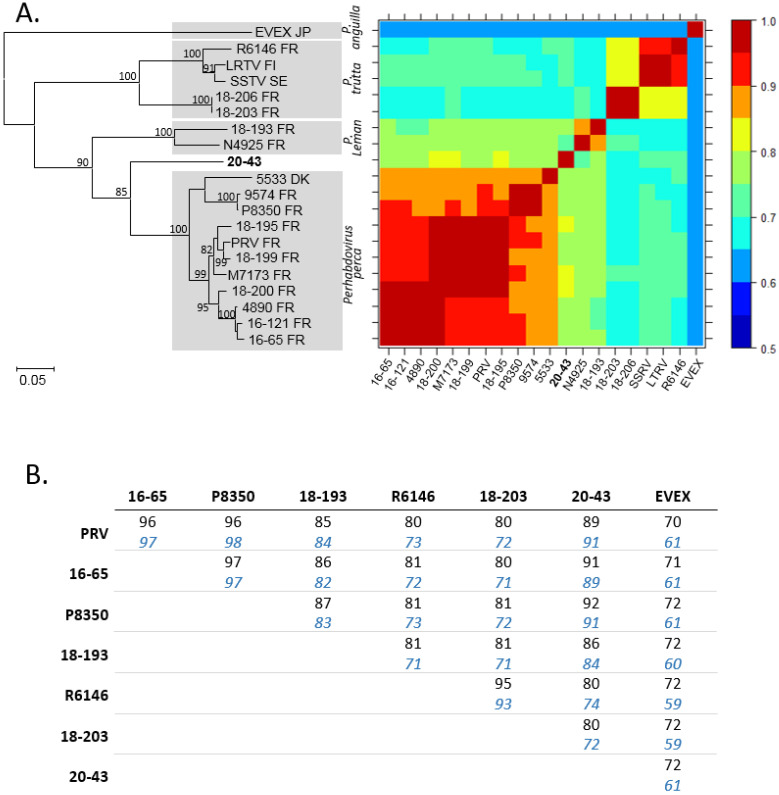
Comparison of selected ORFs of virus 20-43 with homologous sequences of perhabdoviruses. **A.** The maximum-likelihood phylogenetic tree compares the nucleic acid sequences (MEGA7, 1284 positions compared) of the N ORF from a range of viruses of different origins (countries indicated with their two-letter code) belonging to the genus *Perhabdovirus*. EVEX was used as an outgroup. Identities are color-coded in the heat map matrix. **B.** Amino-acid sequence identities (amino-acid sequences) of the L (black) and N (blue italics) ORFs of perhabdoviruses for which both of these sequences are available.

**Figure 4 pathogens-10-01256-f004:**
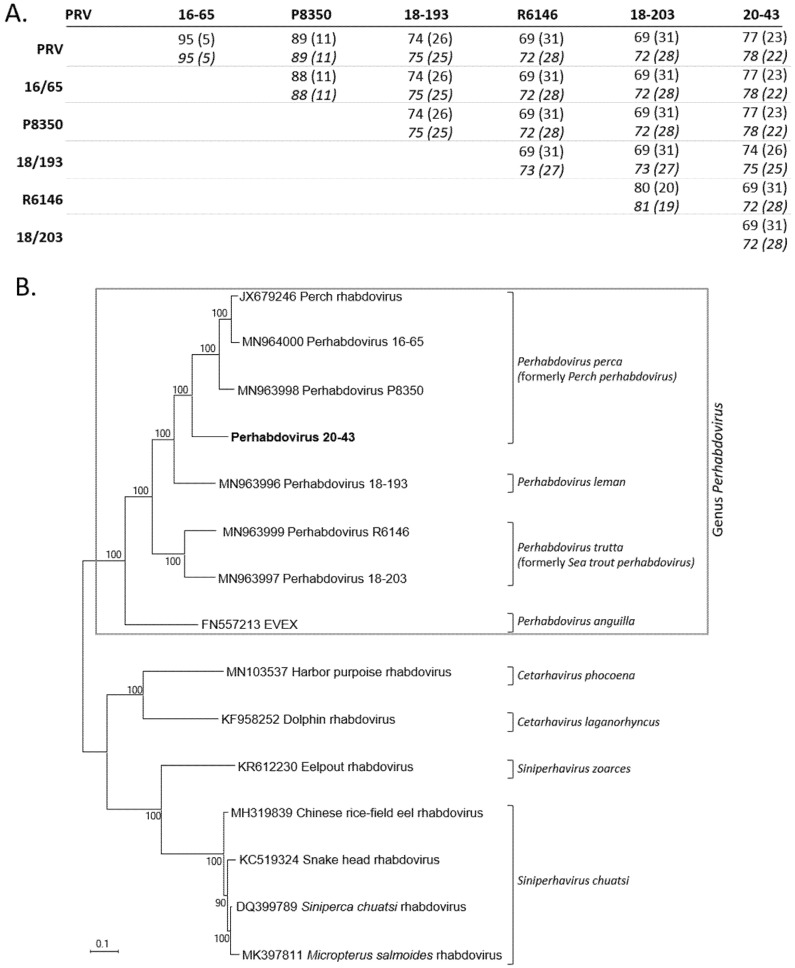
Comparisons and genetic relationships between concatenated homologous ORFs of percid perhabdoviruses and other fish rhabdoviruses. **A**. Pairwise nucleic acid identities in % (divergence in parentheses) of the concatenated ORFs (first line) and the L ORF only (second line, shown in italics). **B**. Maximum-likelihood phylogeny (1000 bootstraps with MEGA10 software) of concatenated ORF sequences (N, P, M, G and L) of virus 20-43 (in bold) and homologous sequences of perhabdoviruses and other fish rhabdoviruses (10107 positions compared). The scale bar indicates genetic distance (substitutions/site).

## Data Availability

The new genetic data are available in GenBank (MW685822).

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
