# Peer review of "A New Lineage of Perch Rhabdovirus Associated with Mortalities of Farmed Perch"

_pathogens, 2021, doi:10.3390/pathogens10101256_

Round 1

Reviewer 1 Report

The author identified a novel isolate of Perhabdovirus perca-like virus, which showing lower sequence homology to the know virus isolates. The manuscript is well written and all information are described.  

Author Response

Thank you for you comment

The author identified a novel isolate of Perhabdovirus perca-like virus, which showing lower
sequence homology to the know virus isolates. The manuscript is well written and all information are
described.

Reviewer 2 Report

The manuscript entitled “A new lineage of Perch Rhabdovirus associated with mortalities of farmed Perch” described the novel genetic lineage of Perhabdovirus affecting farmed Perch in Western Europe.

Some minor mistakes:

Line 128 RT-cPCR

Line 257 any evidence or reference to support this virus could harm tilapia?

Author Response

R2

Some minor mistakes:

Line 128 RT-cPCR

It has been corrected.

Line 257 any evidence or reference to support this virus could harm tilapia?

                There is absolutely no evidence that the present virus 20-43 can infect tilapia. However, for the first time, a rhabdovirus distinct, though related, to perhabdoviruses was shown to be associated with a major outbreak on tilapia in Africa. These results are not published yet. Considering the tremendous importance of tilapia in aquaculture worldwide and the relatively large host range of Perch perhabdovirus, the susceptibility of tilapiines to these viruses should be evaluated.

Reviewer 3 Report

23/8/2021

Pallander and co-authors provide a manuscript with the title "A new lineage of Perch Rhabdovirus associated with mortality of farmed Perch"

Briefly, the authors described isolation of a new virus named 20-43 from perch juveniles dead in an outbreak that occurred at 2019 in Western Europe. The virus has been found related to perhabdovirus. The virus was isolated in tissue culture, detected by conventional PCR and Real time PCR assay. The whole viral genome than was sequenced by a new generation sequencing method, followed by phylogenetic analysis of complete nucleoprotein gene that revealed a new lineage of Perch Rhabdovirus. The methodology used in this study is essentially sound and well excepted.

There are however a few items that should be addressed:

The English language should be reviewed by a native speaker.

Materials and Methods:

  • I would suggest adding a table (Supplementary material) with a list of all the primers used in this study.        Line 90: "We observed a CPE typical of that induced by rhabdoviruses" .Change the syntax of the sentence. Lyssaviruses belong to Rhabdoviridae and do not produce CPE in tissue culture. I would suggest adding a picture or a description of the CPE caused by this virus. Fig 2. - Add the band size of the first conventional PCR picture. (521 bp)
  •  
  •  
  • Results:  
  •  
  •  
  • Add the length of each gene on the NGS picture.Line 186: Add the concatenated length of nucleotide sequences and the genes used.Fig 2: Add the length of the N gene used for phylogenetic tree. Fig 4 B: The concatenated length and genes of homologues ORFs used for the alignment.      
  •  
  •  
  •  
  •  
  •  
  •  
  •  
  •  

Author Response

R3

Pallander and co-authors provide a manuscript with the title "A new lineage of Perch Rhabdovirus associated with mortality of farmed Perch"

Briefly, the authors described isolation of a new virus named 20-43 from perch juveniles dead in an outbreak that occurred at 2019 in Western Europe. The virus has been found related to perhabdovirus. The virus was isolated in tissue culture, detected by conventional PCR and Real time PCR assay. The whole viral genome than was sequenced by a new generation sequencing method, followed by phylogenetic analysis of complete nucleoprotein gene that revealed a new lineage of Perch Rhabdovirus. The methodology used in this study is essentially sound and well excepted.

There are however a few items that should be addressed:

The English language should be reviewed by a native speaker.

Actually, the submitted manuscript was reviewer by a native speaker, remunerated (company SCITEX; Mrs Engel-Gautier). In the revised manuscript, we paid a particular attention to the English.

I would suggest adding a table (Supplementary material) with a list of all the primers used in this study.

A table is now available as a supplementary material (mentioned in M&Ms).

Line 90: "We observed a CPE typical of that induced by rhabdoviruses" .Change the syntax of the sentence.

The sentence has been changed.

Lyssaviruses belong to Rhabdoviridae and do not produce CPE in tissue culture. I would suggest adding a picture or a description of the CPE caused by this virus.

A figure with CPE has been created and is now in the manuscript.

Fig 2. - Add the band size of the first conventional PCR picture. (521 bp)

This detail was added in the figure.

Results:  

Add the length of each gene on the NGS picture.

These information were added on the drawing.

Line 186: Add the concatenated length of nucleotide sequences and the genes used.

The length of the concatenated genes has been added in the text.

Fig 2: Add the length of the N gene used for phylogenetic tree.

This information is now available in the legend of figure 3.

Fig 4 B: The concatenated length and genes of homologues ORFs used for the alignment.    

The length of the concatenated genes has been added in the legend.